# Pathological Myopic Image Analysis with Transfer Learning

**Ruitao Xie**                                   1800261002@email.szu.edu.cn
*College of Information Engineering, Shenzhen University, Shenzhen, China*

**Libo Liu**                                     2016130223@email.szu.edu.cn
*College of Information Engineering, Shenzhen University, Shenzhen, China*

**Jingxin Liu**                                       liujingxin@szu.edu.cn
*College of Information Engineering, Shenzhen University, Shenzhen, China*

**Connor S Qiu**                            connor.qiu12@imperial.ac.uk
*St. Mary's Hospital, Isle of Wight NHS Trust, Isle of Wight, UK*
*Faculty of Medicine, Imperial College London, UK*

## Abstract

We present a summary of transfer learning based methods for several challenging myopic fundus image analysis tasks including classification of pathological and non-pathological myopia, localisation of fovea, and segmentation of optic disc. By adapting existing popular deep learning architectures, our proposed methods have achieved 1st and 2nd place in several tasks at the Pathologic Myopia Challenge (PALM)[1] held at **ISBI2019**.

**Keywords:** Myopia, Fundus Image, Deep Learning, Transfer Learning

## 1. Introduction

According to a recent report by the World Health Organisation (WHO), myopia affects 1.89 billion people worldwide and is likely to affect 2.56 billion people by 2020 (Holden et al., 2016). Myopia has therefore become a global public health issue. Amongst myopic patients, many of them will have high myopia which can lead to pathological myopia, one of the main causes of blindness. Early diagnosis of pathological myopia is therefore very important. With the development of imaging technologies and artificial intelligence methods, computer assisted diagnosis of colour retinal fundus images can help doctors diagnose pathological myopia. However, most of the advances made in this field assume availability of large training datasets. In this paper, we present that transfer learning based methods adapted from existing architectures are able to achieves satisfactory performance on several myopic fundus image analysis tasks including classification, fovea localisation and optic disc segmentation.

## 2. Classification of Pathological and non-Pathological Myopic Images

Distinguishing pathological myopia can be challenging for ophthalmologists. We adopt the ImageNet pretrained ResNet50 (He et al., 2016) with the cross entropy loss function for pathological myopia and non-pathological myopia classification. We augment the 400 images

---

1. https://palm.grand-challenge.org/

training set by adding Gaussian noise($\mu = 0.01, \sigma^2 = [0.01, 0.05]$) and random rotation with 30 degrees. Using such a strategy, the model is able to achieve a very high classification rate. In a pathological and non-pathological myopic fundus image classification challenge held in conjunction with **ISBI2019**, our method has achieved an area under curve (AUC) value of 0.998. In fact, our final algorithm ranked 1st in the final competition.

## 3. Localization of Fovea

The fovea is the optical centre of the eye. Its detection has important clinical implications in early diagnosis for pathological myopia. To detect the position of the fovea in a fundus image, we modified the ImageNet pretrained VGG19 model (Simonyan and Zisserman, 2014) by mixing the features in the 4th and 5th blocks, followed by average Euclidean distance loss function. The schematic of our fovea position localisation network is shown in Figure 1.

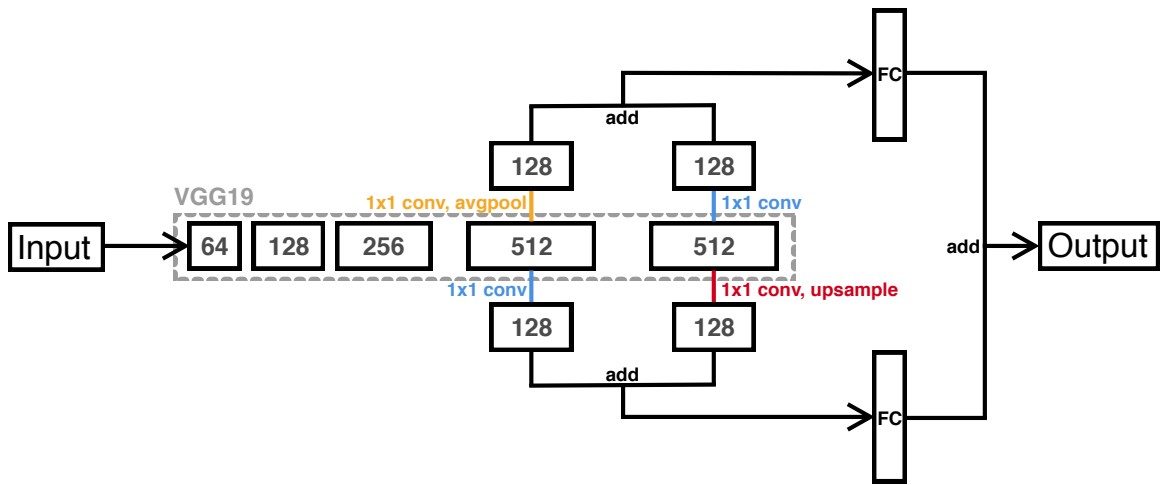

Figure 1: The architecture of modified VGG19 model for fovea localization.

The proposed model achieved 1st place in the fovea localisation task in the PALM challenge. We have also compared our model with standard models such as ResNet50 and VGG19, and results showed that the new network performs better (see Table 1).

Table 1: The Result of Fovea Prediction

|  | **Resnet50** | **Vgg19** | **Our network** |
|---|---|---|---|
| Mean Euclidean Distances (pixel) | 79.98 | 76.28 | 60.65 |
| Euclidean distance Variance (pixel) | 91.31 | 58.33 | 19.15 |

## 4. Segmentation of Disc

The optic disc area is an important part of the fundus image. Segmenting the optic disc can help doctors perform analysis and inform diagnosis. In this work, we propose a U

shape architecture combined ResNet34 (He et al., 2016) and U-Net (Ronneberger et al., 2015). Specifically, the residual module of ResNet34 is utilized as the encoder, followed by a Atrous Spatial Pyramid Pooling (ASPP) (Chen et al., 2018) module. There are three inputs with size of $512 \times 512$, $256 \times 256$, and $128 \times 128$ respectively. The decoder is symmetrical to the encoder with the process of *deconv-concat-conv-conv* in each block, and lateral connections associate encoder and decoder feature maps with the same resolution. The loss function is constrained by both categorical cross-entropy and Dice coefficient.The detail of the segmentation model is illustrated in Figure 2. With this model, we achieved 2nd place in the optic disc segmentation task in the PALM challenge.

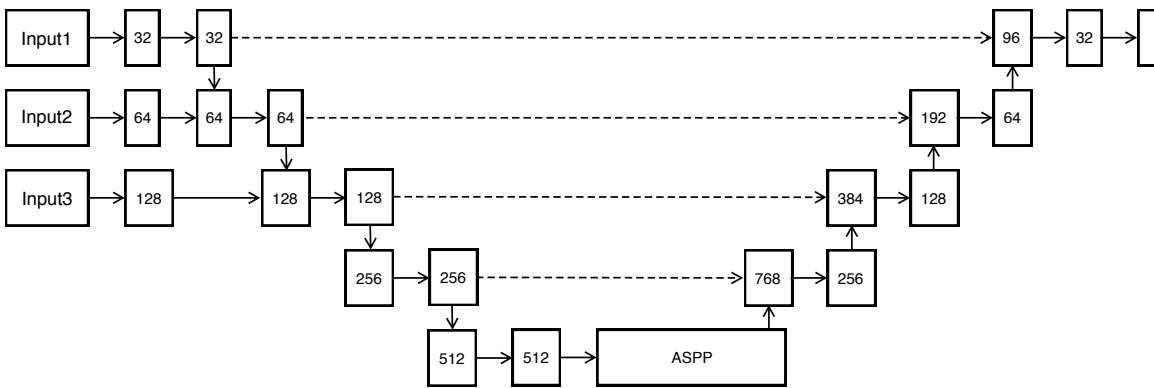

Figure 2: The architecture of the disc segmentation model.

## 5. Concluding Remarks

With the development of deep learning, transfer learning has emerged as a critically important technique. Fine-tuning and modifying existing pretrained models can quickly adapt the success from the nature image domain to the medical image domain (Raghu et al., 2019). In this abstract, we have developed several deep learning based pathological myopia fundus image analysis applications. With pre-trained VGG and ResNet, we can quickly develop several modified models for different tasks, and achieve impressive performance with a small amount of training data. Whilst we have achieved very good results as indicated in the high ranking performances in a very recent pathological myopic fundus image analysis competition, we have also observed some failure cases. More work is needed to fully assess the potential of deep learning in myopic fundus image analysis and its implications in clinical practice.

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
