# OpenReview forum: "Pathological Myopic Image Analysis with Transfer Learning"
_MIDL.io/2019/Conference/Abstract — MIDL Abstract 2019_

### Official Review · AnonReviewer1 · 2019-04-24
**Authors propose methods (some of which are novel) for different tasks involving myopic image analysis. They ranked 1st and 2nd in different tasks (classification, localization, and segmentation) of a grand challenge from ISBI 2019.**

**Rating:** 3
**Confidence:** 3

**Review:**

Summary:
Authors attended in the Pathologic Myopia Challenge (PALM), proposing different architectures for the 3 tasks; (i) classification, (ii) localization, (iii) segmentation. At the point of their participation (or during ISBI, unclear), they have ranked 1st or 2nd place in the above 3 tasks. Authors show that pretraining on natural images (ImageNet) can benefit albeit the different end tasks and input domains.

Strengths:
Authors achieve impressive results in three different challenges and they propose three different methods in this extended abstract (although classification framework seems to be novel only in the data augmentation aspect). The experience of the authors and different approaches they tried tackling these different challenges could flourish useful discussions in MIDL.

Weaknesses:
Authors emphasize on the importance of transfer learning for their high ranking in the different challenges. While finetuning is a valid application of transfer learning, it is not clear that the success of the proposed methods come from the transfer learning. Comparison of the proposed methods with ImageNet pretraining vs training from scratch with only challenge training set would definitely strengthen the claims of the authors.

Major concerns:

- Albeit the manuscript is motivated as myopic image analysis with transfer learning, the benefit of transfer learning is only vaguely constructed, as ranking in the leaderboard. Authors should consider comparing their method also with no-pretraining counterpart.

- It looks like the leaderboard for PALM challenge is still accepting submissions. The results stated by the authors no longer match the current ranks. For the purpose of clarification, please mention the timestamp for when the authors have submitted and/or achieved the stated ranks. In addition, please also state the “model name” the authors used when submitting to the challenge scoreboard.

- Section 4 segmentation proposal does not comment on any pretraining. If there was no pretraining, then I am not sure how does this method qualify as an application of transfer learning.

Minor concerns:

Section 2: please declare the parameters for Gaussian noise and random rotation applied for the purpose of data augmentation.
Section 3: Following the pretrained VGG19 and the randomly initialized additional layers, it is not clear whether VGG19 weights are frozen during the finetuning process for the PALM training set. Please clarify.
Section 3: Figure 1: please state the used activation functions for the layers added above VGG19.
Section 3: Figure 1: Please comment on the “add” operation after the fully connected layers. Are the authors certain that the activations from two FC layers are summed up to regress the row & column of fovea?
Section 3: Please clarify the compared methods in Table 1. Are the compared models also pretrained on ImageNet then finetuned for PALM training set?
Section 3: Table 1: State the unit of the distance metric. In addition, on the leaderboard (http://ai.baidu.com/broad/leaderboard?dataset=pm&task=PM%20Classification), first ranking method has a mean euclidean distance error of 59. I think the authors should stay consistent with the same units as the leaderboard.
Section 3 & 4: The loss function for both methods are not stated.
Section 4: Please declare the full name before writing the abbreviation for ASPP.
Section 5: Authors should elaborate on the failure cases in the recent challenge. Please state the name of the challenge and the particular task where their proposed methods have failed, ideally with also intuition on why it has failed.

Page 2: typo: PLAM -> PALM
Page 2: typo: Section 3 title: fovea -> Fovea
Page 3: typo: Section 4 title: disc -> Disc

---

### Official Review · AnonReviewer2 · 2019-04-30
**Low novelty, little insight on results, but reported high ranking in competition**

**Rating:** 3
**Confidence:** 2

**Review:**

Authors present transfer learning based approaches to myopic image analysis which they report ranked highly (first/second place) in the recent PALM challenge at ISBI 2019. A potential concern with this is that, according to instructions reviewers received, previous conference presentation should not be acceptable for MIDL extended abstracts. However, as far as I could find out, no description of the algorithm was published at the challenge, not even in abstract form. Therefore, in my opinion, re-submission to MIDL extended abstracts should be fine. The submission is within scope, and the reported high ranking should make it relevant despite several shortcomings:

 - The main claimed benefit is transfer learning, but the extended abstract contains no information on how much the results actually gained from transfer learning (as opposed to learning with the same architecture, same data, same augmentation from scratch)

 - The authors state that they encountered important failure cases, but do not show or discuss them. In fact, not a single result from the reported networks is shown.

 - Technical novelty is limited, used networks are rather straightforward variations of well-tested ones.

 - What are the units in Table 1?

Overall, I think this submission is very much at the borderline for being acceptable, and I would not champion it if other reviewers should argue against it.

---

### Decision · Program_Chairs · 2019-05-06
**Acceptance Decision**

Accept